# The radiative impact of out-of-cloud aerosol hygroscopic growth during the summer monsoon in southern West Africa

Sophie L. Haslett[1,a], Jonathan W. Taylor[1], Konrad Deetz[2], Bernhard Vogel[2], Karmen Babić[2], Norbert Kalthoff[2], Andreas Wieser[2], Cheikh Dione[3], Fabienne Lohou[3], Joel Brito[4], Régis Dupuy[4], Alfons Schwarzenboeck[4], Paul Zieger[5,6], and Hugh Coe[1]

[1]School of Earth and Environmental Sciences, University of Manchester, Manchester, United Kingdom
[2]Institute of Meteorology and Climate Research, Karlsruhe Institute of Technology (KIT), Karlsruhe, Germany
[3]Laboratoire d'Aérologie, Université Paul Sabatier Toulouse III (UPS), Toulouse, France
[4]Laboratoire de Météorologie Physique, Université Clermont Auvergne, Aubière, France
[5]Department of Environmental Science and Analytical Chemistry, Stockholm University, Stockholm, Sweden
[6]Bolin Centre for Climate Research, Stockholm University, Stockholm, Sweden
[a]Now at: Department of Environmental Science and Analytical Chemistry, Stockholm University, Stockholm, Sweden

*Correspondence to:* Hugh Coe (hugh.coe@manchester.ac.uk)

**Abstract.**

Water in the atmosphere can exist in the solid, liquid or gas phase. At high humidities, if the aerosol population remains constant, more water vapour will condense onto the particles and cause them to swell, sometimes up to several times their original size. This significant change in size and chemical composition is termed hygroscopic growth and alters a particle's optical properties. Even in unsaturated conditions, this can change the aerosol direct effect, for example by increasing the extinction of incoming sunlight. This can have an impact on a region's energy balance and affect visibility. Here, aerosol and relative humidity measurements collected from aircraft and radiosondes during the Dynamics-Aerosol-Chemistry-Cloud Interactions in West Africa (DACCIWA) campaign were used to estimate the effect of highly humid layers of air on aerosol optical properties during the monsoon season in southern West Africa. The effects of hygroscopic growth in this region are of particular interest due to the regular occurrence of high humidity and the high levels of pollution in the region. The Zdanovskii, Stokes and Robinson (ZSR) mixing rule is used to estimate the hygroscopic growth of particles under different conditions based on chemical composition. These results are used to estimate the aerosol optical depth (AOD) at $\lambda = 525$ nm for 63 relative humidity profiles. The median AOD in the region from these calculations was 0.36, the same as that measured by sun photometers at the ground site. The spread in the calculated AODs was less than the spread from the sun photometer measurements. In both cases, values above 0.5 were seen predominantly in the mornings and corresponded with high humidities. Observations of modest variations in aerosol load and composition are unable to explain the high and variable AODs observed using sun photometers, which can only be recreated by accounting for the very elevated and variable RHs in the boundary layer. Most importantly, the highest AODs present in the mornings are not possible without the presence of high RH in excess of 95 %. Humid layers are found to have the most significant impact on AOD when they reach RH greater than 98 %, which can result in a wet AOD more than 1.8 times the dry AOD. Unsaturated humid layers were found to reach these high levels of RH in 37 % of observed cases. It can therefore be concluded that the high AODs present across the region are driven by the high humidities, and are

then moderated by changes in aerosol abundance. Aerosol concentrations in southern West African are projected to increase substantially in the coming years; results presented here show that the presence of highly humid layers in the region is likely to enhance the consequent effect on AOD significantly.

## 1 Introduction

Aerosol particles can absorb water even in sub-saturated conditions. In humid environments, this process can increase their size substantially. This increase in the aerosol liquid water content affects a particle's interactions with radiation by increasing its diameter and changing its refractive index. Therefore, increased aerosol water content can alter the aerosol direct effect on radiative forcing. In addition, these alterations to aerosol characteristics can affect visibility and form haze or alter the chemical interactions of particles in the atmosphere (Chen et al., 2012).

The extent to which an aerosol particle takes on water (its hygroscopic growth) is dependent on a number of variables, including its chemical composition, the ambient aerosol concentration and the relative humidity (RH) in its environment. Due to the complexity of this function, hygroscopic growth has been identified as one of the key uncertainties in aerosol radiative forcing (Forster et al., 2007). While the growth of inorganic aerosol species is reasonably well understood, the complexity of organic aerosols, which can include hundreds of different compounds, has made it difficult to establish a definitive approach to investigating its hygroscopic properties in a mechanistic way (Gysel et al., 2007; Topping et al., 2005). A number of studies have been carried out, to examine hygroscopic growth empirically in the ambient environment (e.g. Esteve et al., 2014; Gysel et al., 2007; Hersey et al., 2009; Highwood et al., 2012; Kamilli et al., 2014; Liu et al., 2011).

A particle's hygroscopic properties can be described using kappa ($\kappa$), which was developed by Petters and Kreidenweis (2007) to collate all of the chemical variables that affect hygroscopicity into a single parameter. Although this value can vary slightly with RH and temperature, it is robust enough to be considered a constant for a given chemical composition in most ambient cases. This value is closely related to an aerosol's hygroscopic growth factor (HGF) in a given environment - the ratio between its diameter after it has absorbed water and its dry diameter. These quantities present challenges for measurement, as the majority of aerosol instruments dry aerosol particles before measuring their properties.

Some studies have used the Hygroscopic Tandem Differential Mobility Analyser (H-TDMA) to measure the wet diameters of aerosol particles (Swietlicki et al., 2008), which makes a direct calculation of the HGF possible above relative humidities of around 90 %. A particle's hygroscopicity depends on its chemical constituents. Therefore, another approach is to estimate aerosol water content based purely on measurements of chemical composition and on an aerosol population's size distribution. If an aerosol population with different chemical constituents can be assumed to be internally mixed, the Zdanovskii, Stokes and Robinson (ZSR) mixing rule (Stokes and Robinson, 1966; Zdanovskii, 1948) allows the HGF of a mixed particle to be estimated based on the known growth factors of the pure constituents. While ambient aerosol is known to exist in a combina-

tion of internal and external mixing states, this assumption is reasonable for more aged, regional aerosol populations (Boucher et al., 2013; Pratt and Prather, 2010). When the HGF is known, the size distributions and known densities of aerosol particles and consituents can be used to estimate the volume of water in an aerosol sample. A number of closure studies have been carried out to assess the reliability of these calculations compared with H-TDMA measurements, with compelling results (Gysel et al., 2007; Hersey et al., 2009; McFiggans et al., 2005). Both McFiggans et al. (2005) and Gysel et al. (2007) found closure was not possible when nitrate loadings were included in calculations. Gysel et al. (2007) suggested that this was likely due to the evaporation of nitrate in the H-TDMA, although some studies have found closure between ZSR calculated and measured hygroscopic growth for nitrate-containing particles (Guo et al., 2015; Henningan et al., 2015). Another explanation for found discrepancies in optical closure studies using H-TDMAs is that these instruments usually only select dry sizes in the lower submicron size range and thus miss optically important size ranges in the accumulation and coarse mode (Zieger et al., 2011).

A number of recurring features have been observed in various HGF measurements: Aklilu et al. (2006) noted that a high proportion of sulphate is often correlated with high HGF values. This is related to the highly hygroscopic nature of sulphate-containing aerosols compared with other species. Studies have shown that organic aerosol does grow hygroscopically, though to a lesser extent than many inorganic species (e.g. Nguyen et al., 2016). The HGF is slightly larger for secondary organics and more aged aerosol due to its high oxidation levels. However, the influence of even highly-oxidised organic compounds is small compared with that of inorganic compounds. Therefore, the organic to inorganic ratio has been found to be a more influential factor for the HGF than the composition of the organic aerosol itself (McFiggans et al., 2005). A single value is therefore often used to represent the HGF of organic aerosol. Urban environments have been found to include several types of aerosol that are primarily hydrophobic (they do not readily absorb water). These include freshly emitted combustion particles due to their high soot content, and insoluble organic compounds (Swietlicki et al., 2008). The HGF correlates closely with the RH in a region, and has been found to follow diurnal patterns in RH closely (e.g. Liu et al., 2011).

When the aerosol water content in a region has been calculated, this value can be used in tandem with the calculated refractive index of wet particles to calculate changes to the scattering and extinction properties of the particles, for example using a framework like that described by Esteve et al. (2014). Thus, changes to the radiative properties of particles and their consequent impact on radiative forcing can be estimated.

During the monsoon season in West Africa, the RH is often high, which results in substantial aerosol hygroscopic growth. Therefore, this is a region in which hygroscopic growth is likely to have a large impact on radiative forcing. This supposition is supported by a previous study in southern West Africa, which found a strong increase in backscatter in the region between 18:00 and 00:00 UTC. This was thought to be related to aerosol growth due to increases in humidity during these times (Babić et al., 2018). Anthropogenic emissions in the region are projected to increase substantially by 2030 (Liousse et al., 2014), which will increase the impact of this hygroscopic growth.

This study presents radiosonde and aircraft data from the Dynamics-Aerosol-Chemistry-Cloud Interactions in West Africa (DACCIWA) campaign, a large field campaign that took place in southern West Africa during June and July 2016 (Knippertz et al., 2015, 2017; Flamant et al., 2018a). Chemical and physical aerosol measurements from the British Antarctic Survey (BAS) Twin Otter and the French Service des Avions Français Instrumentés pour la Recherche en Environnement (SAFIRE) ATR-42 aircraft are presented here. The ZSR technique has been used to estimate the HGF and aerosol liquid water content

based on chemical measurements.

## 2   Data and Methods

Data were obtained during the DACCIWA aircraft and ground-based campaigns in southern West Africa during June and July 2016 (Flamant et al., 2018a; Kalthoff et al., 2018). The chemical data presented here were collected by the British Antarctic

Survey Twin Otter aircraft, from 11 flights carried out between 6 July and 15 July. The Twin Otter's operating area during the campaign was from 0.6 °W to 2.7 °E and from 5.4 °N to 8.0 °N. During DACCIWA, airborne measurements were made of aerosol properties with a focus on both regional aerosol properties and the effects of urban emissions. Flight patterns consisted of a series of profiles, and straight and level runs at different altitudes. The focus of the Twin Otter aircraft was on low-tropospheric measurements, below 3 km. Instrumentation on board the aircraft measured submicron aerosol chemical

composition, optical properties and physical properties.

An Aerodyne Compact Time-of-Flight Aerosol Mass Spectrometer (AMS; Aerodyne Research Inc., Billerica, MA, USA) (see Canagaratna et al., 2007; Drewnick et al., 2005) was used to measure the chemical composition of the submicron non-refractory aerosol mass. Measurements of organics, nitrate, sulphate and ammonium were collected; chloride concentrations

were found to be consistently below the instrument's detection limit and so are not included here. The ionisation efficiency of the instrument was calibrated several times during the campaign using ammonium nitrate and once with ammonium sulphate. A collection efficiency of 0.5 was applied to the data, which is typical for ambient data (Middlebrook et al., 2011). Comparisons between the AMS and the volume convolved size distribution show that other components, for example mineral dust, contribute little to the submicron aerosol mass and that coarse mode concentrations were low. While mineral dust makes an

important contribution to West African aerosol throughout much of the year (Ji et al., 2018), its concentration in the southern region during the monsoon season is often low due to the predominant southerly monsoon winds (Knippertz et al., 2017). Instances of lofted dust layers have been observed to the north of the West African region, but have not been considered in this analysis; if present, they would simply add to the total AOD in the column. Black carbon concentrations were established using a Single Particle Soot Photometer (SP2; Droplet Measurement Technologies, Longmont, CO, USA). This produced a

time series of the mass concentration of refractory black carbon larger than $0.3\,\mathrm{fg}$. Calibrations using Aquadag were carried out several times during the campaign, with the standard scaling factor of 0.75 applied to the calibration curve (Laborde et al., 2012). Particle size distributions were measured by a Scanning Mobility Particle Sizer (SMPS; TSI Inc.) on board the ATR-42

aircraft. This aircraft carried out DACCIWA flights during the same time period as the Twin Otter, including a significant number in the region of interest here (Brito et al., 2018). The SMPS determines particle sizes using electrical mobility, scanning through the diameter size range from 0.02 - 0.5 $\mu$m. This produces an aerosol size distribution every 120 s.

Observations were considered alongside RH profiles from a number of radiosonde soundings from the DACCIWA supersite in Savè (8.03 °N, 2.48 °E). Radiosonde releases were carried out from the Savè supersite for a period of two months (June and July), with several releases being made each day. This allowed the compilation of a robust set of RH statistics. The setup of measurements and the overview of the diurnal cycle of the atmospheric boundary layer conditions from the Savè ground site are presented in more detail by Kalthoff et al. (2018).

In order to ensure that aircraft data were comparable with RH readings in this region, aircraft data have been included here if collected further north than 6.4 °N and further east than 1.3 °E, as shown by the box in Fig. 1. The chemical aerosol data collected in this region were found to be reasonably invariant across the campaign period (medians and interquartile ranges are displayed in Fig. 2); thus, these data have been averaged in order to increase the robustness of results. As observations of RH were taken from the Savè ground site, results shown here apply to the atmosphere above Savè. The Twin Otter flight tracks from the campaign are displayed in Fig. 1.

## 2.1  $\kappa$-Köhler theory and the ZSR mixing rule

The single-parameter approach to describing a particle's propensity to grow hygroscopically proposed by Petters and Kreidenweis (2007) was employed here. This combines all variables that depend on a particle's chemical composition into a single parameter, $\kappa$. The relationship between the $\kappa$-value and the HGF is expressed by Eq. 1:

$$\frac{\text{RH}}{exp(\frac{A}{D_d \text{HGF}})} = \frac{\text{HGF}^3 - 1}{\text{HGF}^3 - (1 - \kappa)} \tag{1}$$

where $RH$ is the relative humidity expressed as a fraction, $HGF$ is the hygroscopic growth factor, $D_d$ is the dry particle diameter and

$$A = \frac{4\sigma_{s/a} M_w}{RT\rho_w} \tag{2}$$

where $\sigma_{s/a}$ is the surface tension at the solution-air interface, $M_w$ is the molar mass of water, $R$ is the universal gas constant, $T$ is the absolute temperature and $\rho_w$ is the density of water. This equation is solved iteratively.

The value of $\kappa$ increases with increasing hygroscopicity. Thus, $\kappa$ will be 0 for a non-hygroscopic particle, above 1 for more hygroscopic particles and up to around 1.5 for the most hygroscopic particles such as NaCl. Typically, values for continental ambient aerosol fall between 0.1 and 0.4 (Pringle et al., 2010). Values of $\kappa$ used for the species considered here are listed in

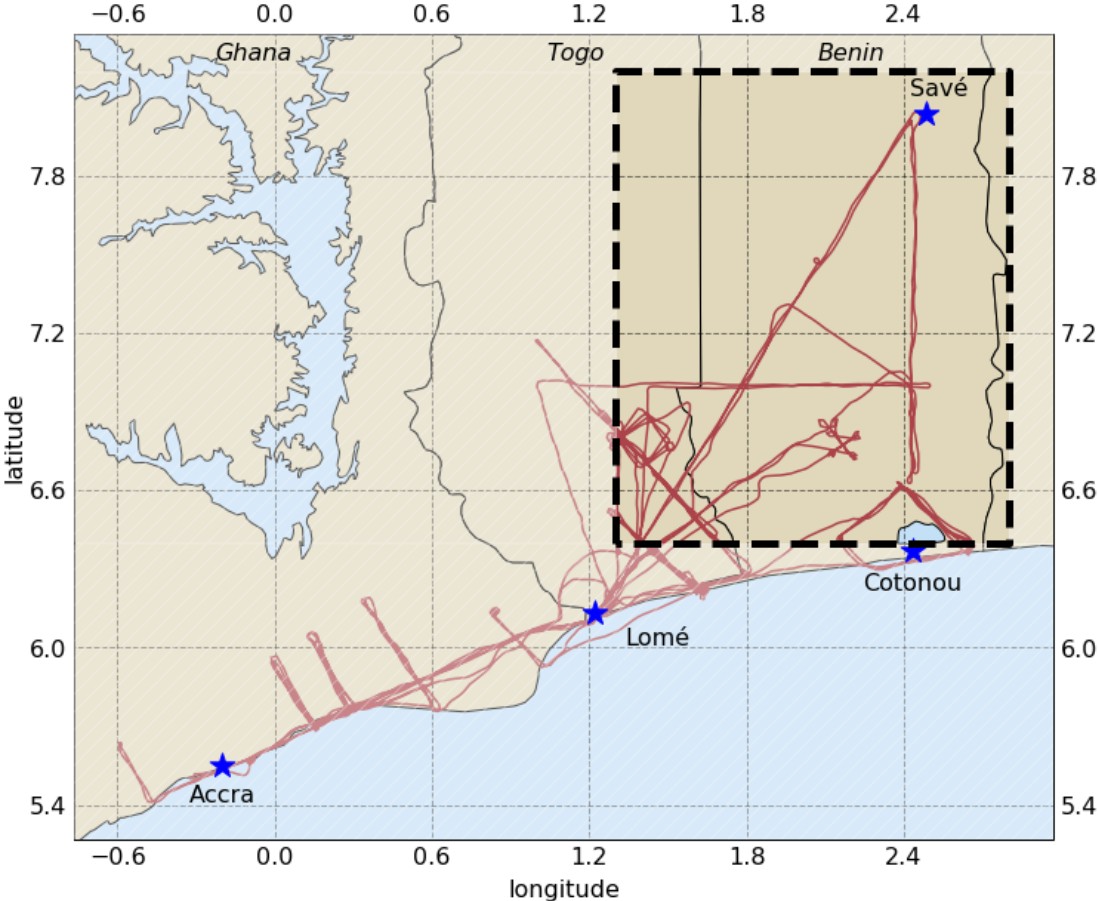

**Figure 1.** Map of the Twin Otter flight paths. The box represents locations where data was used in this analysis to compare with radiosonde data from the Savè supersite.

table 1. If ambient aerosol is assumed to be internally mixed, the $\kappa$ values of the different components can be combined to provide a total $\kappa$ value for the mixed particle, using Eq. 3:

$$\kappa = \sum_i \varepsilon_i \kappa_i,$$
(3)

where $\varepsilon_i$ represents the volume fraction of an individual chemical component. This follows from the ZSR mixing rule described by Zdanovskii (1948) and Stokes and Robinson (1966). Once the value of $\kappa$ has been determined for a given particle and if the RH is known, the relationship shown in Eq. 1 can be used to calculate the particle's expected HGF.

**Table 1.** Density, kappa values and refractive indices for aerosol species considered here.

| Species | Density ($kg\,m^{-3}$) | $\kappa$ | Refractive index ($\lambda$ = 525 nm) |
|---|---|---|---|
| $NH_4NO_3$ | 1725[a] | 0.68[f] | $1.6 - 0i$[a] |
| $(NH_4)_2SO_4$ | 1769[b] | 0.52[f] | $1.53 - 0i$[i] |
| $NH_4HSO_4$ | 1780[c] | 0.56[f] | $1.47 - 0i$[c] |
| Organics | 1200[d] | 0.1[g] | $1.46 - 0.021i$[d] |
| Black carbon | 1800[e] | 0[h] | $1.85 - 0.79i$[j] |

[a] Weast (1985), in Morgan et al. (2010)

[b] Penner et al. (1998), in Morgan et al. (2010)

[c] Lowenthal et al. (2003)

[d] Stelson (1990), in Taylor et al. (2015)

[e] Bond and Bergstrom (2006), in Morgan et al. (2010)

[f] Liu et al. (2014)

[g] Suda et al. (2012)

[h] Weingartner et al. (1997), in Jurányi et al. (2010)

[i] Toon et al. (1976), in Morgan et al. (2010)

[j] Bond and Bergstrom (2006)

## 2.2 Ion pairing

The AMS provides a time series of nitrate ($NO_3^-$), sulphate ($SO_4^{2-}$) and ammonium ($NH_4^+$) fragments. However, quantities of complete neutral salts are needed in order for the ZSR mixing rule described above to be used. Here, moles of neutral salts were established using the ion pairing scheme outlined by Gysel et al. (2007), as shown in Eq. 4. Calculations for $H_2SO_4$ and $HNO_3$ are not included as the aerosol was found to be in charge balance in all cases considered here; the ZSR calculations returned zero for both of these compounds.

$$
\begin{aligned}
n_{NH_4NO_3} &= n_{NO_3^-} \\
n_{NH_4NSO_4} &= \min(2n_{SO_4^{2-}} - n_{NH_4^+} + n_{NO_3^-}, n_{NH_4^+} - n_{NO_3^-}) \\
n_{(NH_4)_2SO_4} &= \max(n_{NH_4^+} - n_{NO_3^-} - n_{SO_4^{2-}}, 0)
\end{aligned}
\tag{4}
$$

## 2.3 Optical properties

Aerosol optical properties were calculated using a Mie code, based on a set of algorithms developed by Yang (2003), which uses a supplied refractive index, wavelength of light and aerosol diameter to calculate properties including the extinction coefficient for a multi-layered sphere. The refractive index used here was calculated by considering the weighted sum of the refractive indices of the individual chemical compounds in a particle, again assuming volume mixing. This technique does not account for some effects, including possible lensing effects from the addition of water to particles or the co-condensation of

other volatile compounds with water. The individual refractive indices of the components considered are shown in table 1. The effect of absorbed water on the refractive indices was taken into account, with the refractive index for water of $1.33 - 0i$ being used.

## 3 Results

### 3.1 Observations

Figure 2a shows the average chemical composition of submicron aerosol in the region outlined by the box in Fig. 1. Data have been averaged to 500 m bins. Figure 2b shows the relative proportions of each chemical species at each altitude bin. As can be seen from Fig. 2b, the chemical distribution of aerosol was reasonably constant in this region in the lower five bins, with the overall concentration decreasing with altitude. Organic aerosol contributed the largest proportion to the aerosol loading, making up over 50 % of the total mass. $SO_4{}^{2-}$ was around 20 % in all cases. Black carbon was just under 15 %. Contributions from $NO_3{}^-$ and $NH_4{}^+$ were each less than 10 %. This proportional composition changes in the 3000 m bin, with $SO_4{}^{2-}$ becoming the most prominent and the $NO_3{}^-$ contribution decreasing. it is likely this reflects the composition of aerosol immediately above the boundary layer top. However, since this altitude was only flown once during the morning of 15 July, it cannot be determined how representative this measurement is of the composition in general. If this were to be included in the profile it would only enhance the extinction at the top of the boundary layer, although the absolute concentrations are very small and make a limited contribution to the total ambient AOD.

Figure 3 shows the normalised average number and surface area size distributions of submicron aerosol in the region outlined by the box in Fig. 1, as measured by the SMPS on board the ATR aircraft. Two distinct size modes can be seen, indicating that the region contains smaller, fresher aerosol with a mode at around 60 nm and a large, accumulation mode containing more aged aerosol. The smaller, Aitken mode was highly variable, as can be seen by the large spread in the data in Fig. 3a. The accumulation mode, however, remained present and relatively stable (with an interquartile range of $\pm$ 30 % of the median) throughout the campaign. An optical particle counter (Grimm, model 1.109) was present on the Twin Otter aircraft during the campaign and did not measure a significant coarse mode during these flights. We therefore limit this analysis to accumulation mode aerosol only and use the extrapolated measured SMPS distribution for the remainder of this work.

Aircraft observations during the DACCIWA campaign period showed little variation in total aerosol concentration as a result of the diurnal cycle (see Fig. 2). When the data were sorted according to the time of day they were observed and considered in two-hourly time bins, the largest deviation in the bin means from the overall campaign mean was around 11%. Comparing aerosol profiles from different times of day did not reveal any up- or downwards trend in the aerosol concentrations as a function of the time of day, although there was some variability in mass loadings from one day to the next. Similarly, there was no evidence in the data for a trend in the proportional distribution of aerosol species across the diurnal cycle. For example,

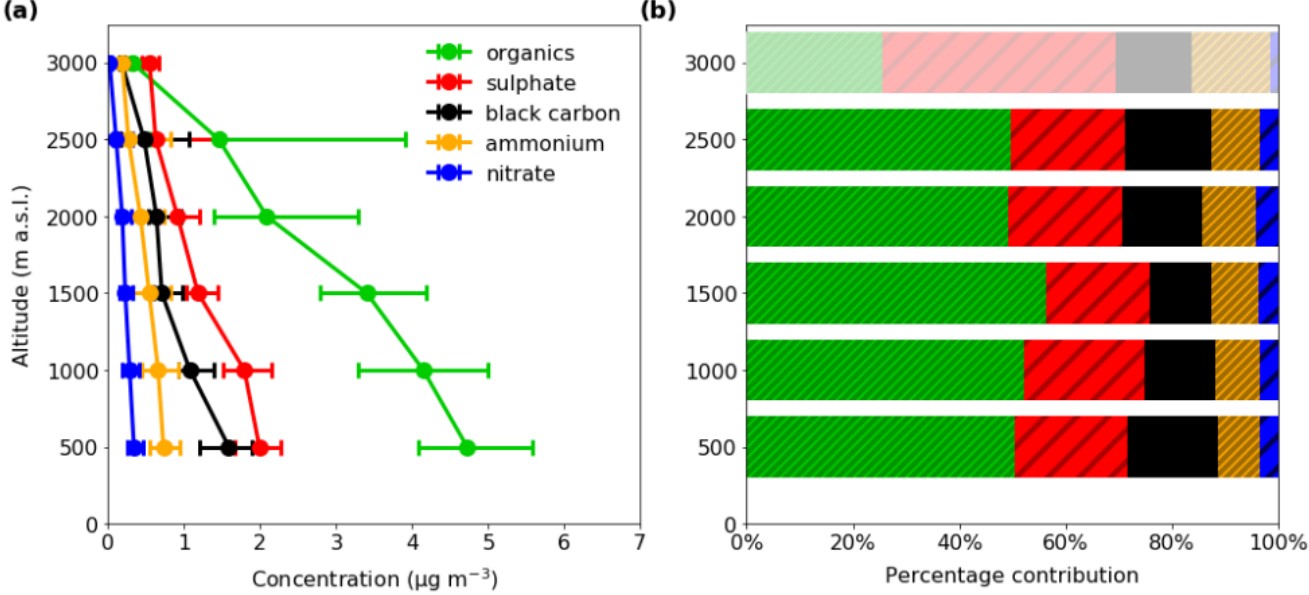

**Figure 2.** **(a)** Average chemical composition of aerosol in the Savè region, as measured by the AMS and SP2 (black carbon). Results are displayed at standard temperature and pressure (STP) and altitude is above sea level (a.s.l.) The centre point indicates the median concentration and the bars the lower and upper quartiles. **(b)** The percentage contribution of each chemical species in each altitude bin. The 3000 m is shaded as data only exist at this altitude from one hour on 15th July.

the average organic to sulphate ratio varied by less than 10 % across the diurnal cycle. The boundary layer aerosol during the DACCIWA campaign was found to have been significantly influenced by long-distance transport from biomass burning in cen-
tral and southern Africa, which is likely to account for the homogeneity in aerosol properties observed here. This phenomenon will be explored in future work. All DACCIWA flights took place between the hours of 07:00 and 19:00 and were therefore unable to provide information about potential changes in aerosol properties occurring during the night.

As the objective of this work is to examine the effects of high RH, and given the lack of sufficient evidence in this dataset
suggesting a trend in aerosol concentrations over the diurnal cycle, a median aerosol profile and chemical composition are used here to explore the effects of changes in RH on the AOD. However, it is important to note that the resultant calculations only provide an indication of the effect of RH on the AOD and are unable to capture the true range of AOD that would result from including variations in the aerosol loading, which would scale the resulting AOD linearly.

Figure 4 shows RH measurements from radiosondes released from the ground site in Savè. These plots show the mean RH and standard deviations at every 100 m interval up to 3 km, averaged across four times of day. These data show a clear diurnal cycle in the RH, with a humid boundary layer during the night and early morning becoming drier towards noon and into the

evening. These features are characteristic of the Gulf of Guinea maritime inflow (Adler et al., 2017; Deetz et al., 2018b), a coastal air mass that propagates inland in the evening, bringing cool air to the southern West African region. The increase of
RH in the evening is mainly caused by cooling due to this cold air advection (Adler et al., 2018; Babić et al., 2018). The RH in the pre-frontal region is generally around 75 %, while post-frontal RH rises to be almost consistently above 90 % (Deetz et al., 2018a) and leads to low-level cloud formation in the night (Adler et al., 2018; Babić et al., 2018).

The dataset of RH measurements used here is valuable, as models are often unable to replicate the vertical RH profile
(Hannak et al., 2017). This is due to the complex interplay between shallow and deep convection, which changes the vertical distribution of water vapour and clouds. These factors influence the surface energy budget, which can in turn create deep convection, in a feedback loop. The uptake of water by aerosol particles adds another layer of complexity to this picture.

The time series panel gives insight into the day-to-day variability in RH. Four phases of the 2016 West African monsoon
have been defined by Knippertz et al. (2017) and are delineated by the vertical dashed lines in Fig. 4e. The first, pre-onset phase was characterised by rainfall near the coast, which moved inland during the second, post-onset phase. This can be seen in the drier start to the period shown in Fig. 4e. This was followed by a period of higher humidity in phase 2, during which humidities rarely dropped below 80 %. The final week of the post-onset phase was slightly drier than those preceding it, which was associated with a cyclonic system that passed Savè on 16 July, transporting dry air northwards (Kalthoff et al., 2018).
During phase 3, the rainfall maximum shifted again back to the coast and during phase 4, the recovery of the monsoon, the maximum was once again found inland.

Using an average aerosol particle size of 200 nm (as measured by the SMPS) and assuming an aerosol composition of 52 % organic, 21 % sulphate, 14 % black carbon, 9 % ammonium and 4 % nitrate (see Fig. 2b), it was possible to estimate the average
HGF of particles at each of the altitudes and times of day in Fig. 4 using the method described in sections 2.1 and 2.2. The value of $\kappa$ for this composition was calculated using the method described in Section 2.1, assuming the composition to be the same across the size distribution. This produced a value of $\kappa = 0.22$, which is typical for ambient aerosol (Pringle et al., 2010). These results are displayed in Fig. 5. The plots show the extent to which the diameter of an average particle would be expected to grow under the given conditions. The shape of the HGF profiles is governed by the RH statistics shown in Fig. 4. It can be
seen that the sensitivity to changes in RH is more pronounced where the humidity is high. In drier regions, for example at low altitudes at 18 UTC, reasonably significant variation in RH produces very little change in particles' hygroscopic growth. This suggests that even thin layers of high RH are likely to have a disproportionately large impact on the radiative properties of the column. The particles are larger due to the high RH, and these larger particles scatter sunlight more effectively.

Ceilometer observations from the DACCIWA ground site in Savè during the campaign have been explored by Babić et al. (2018). This instrument uses a laser to determine atmospheric backscatter, which allows the determination of cloud base. A strong increase in backscatter was noted between 18:00 and 00:00 UTC, which is thought to be related to aerosol hygroscopic

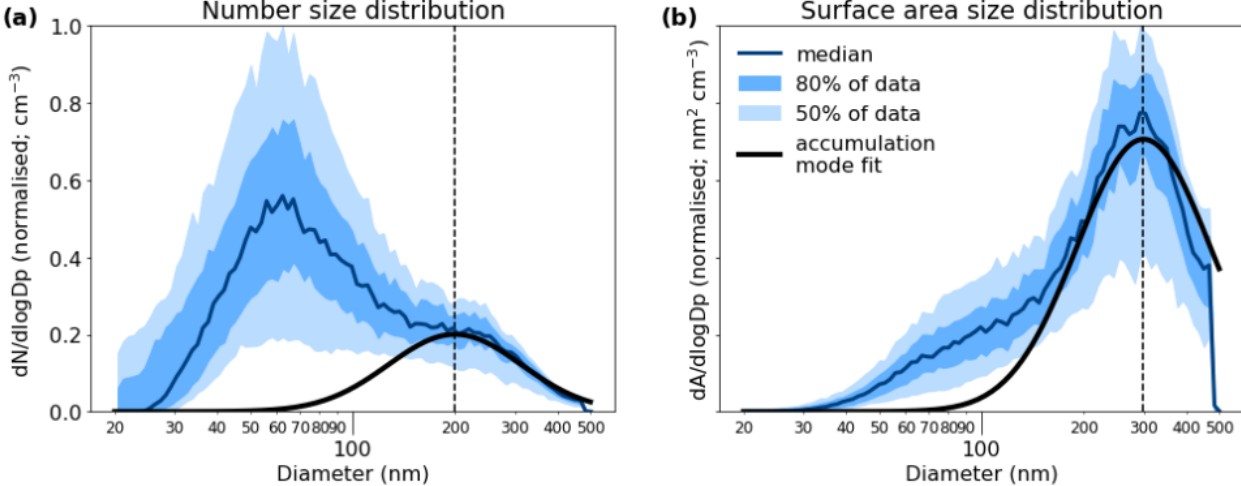

**Figure 3.** The **(a)** number size distribution and **(b)** volume size distribution of submicron aerosol below 2500 m in the area outlined by the box in Fig. 1. The dark blue lines show the median size distributions; the darker shading represents data between the $25^{th}$ and $75^{th}$ percentiles (50 % of the data) and the lighter shading data between the $10^{th}$ and $90^{th}$ percentiles (80 % of the data). The thick black line represents a lognormal curve fitted to the accumulation mode. The rapid drop-off at the high end of the surface area distribution is an artefact of the SMPS size limit.

growth. The frequently high HGF layers shown here are consistent with this interpretation.

The volume of absorbed water at high humidities can make a significant difference to the composition and properties of aerosol particles. Figure 6a shows the percentage contribution of the compounds being considered here to the particle volume at different relative humidities. Above around 80 % RH, water contributes over half of the volume and above around 95 % RH, water contributes more than 80 % to the volume of an individual particle, both growing the particle substantially and changing its optical properties. The decrease in aerosol refractive index as the particles grow is illustrated in Fig. 6b.


### 3.2   Calculating aerosol optical depths

In order to establish the impact of humid layers on column optical properties, the sub-3 km aerosol optical depth (AOD) was calculated for each of the 63 unsaturated radiosonde profiles taken from the Savè ground site during the DACCIWA campaign. The steps of the approach used to establish the AOD are outlined below using a case study RH profile from 8 July at 1057 UTC
(local time = UTC+1 hr). This approach was then applied to all RH profiles in the dataset.

   1.  Establish the average aerosol profile.

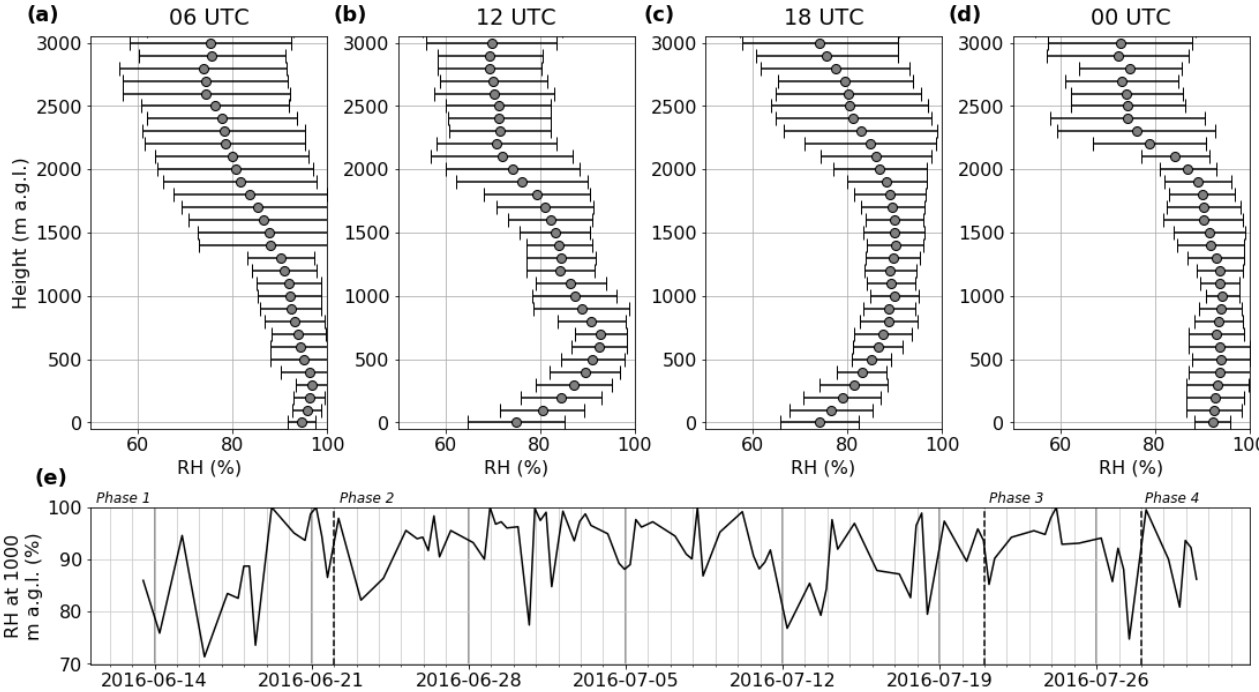

**Figure 4.** The mean and standard deviation of relative humidity measurements with altitude at **(a)** 06 UTC (n=46), **(b)** 12 UTC (n=15), **(c)** 18 UTC (n=20) and **(d)** 00 UTC (n=15), from radiosondes launched from the Savè ground site. Altitudes are above ground level (a.g.l.) **(e)** A time series of the RH at 1000 m. Dashed vertical lines indicate the four different monsoon phases, as defined by Knippertz et al. (2017). (NB: Local time in Benin is UTC+1hr.)

The median aerosol mass concentration in the area of interest was calculated for each 100 m bin from 300 m to 3000 m. Data were not collected from lower than this within this region, so an average datapoint from further south was used here to represent the 200 m bin. The resulting median profile can be seen in Fig. 7, alongside the interquartile range and 10$^{th}$ to 90$^{th}$ percentiles and the number of flights providing data in each bin. The median profile was used to calculate the AOD for all of the radiosonde profiles.

2. Establish an idealised aerosol size distribution at each altitude.

Figure 3 shows the average aerosol size distribution at different altitudes, with two clear modes (Aitken and accumulation mode) being identified. In terms of aerosol hygroscopicity and changes to optical properties, the larger mode is the most significant as it contains the vast majority of the aerosol surface area and is of a size comparable with incoming radiation, making it the more optically active size range. The number distribution is dominated by the small, fresh Aitken mode aerosol, which shows a large degree of variation but contributes little to the overall optical properties of the aerosol; therefore, the accumulation mode was isolated here to calculate optical effects. It can be noted from Fig. 3 that there

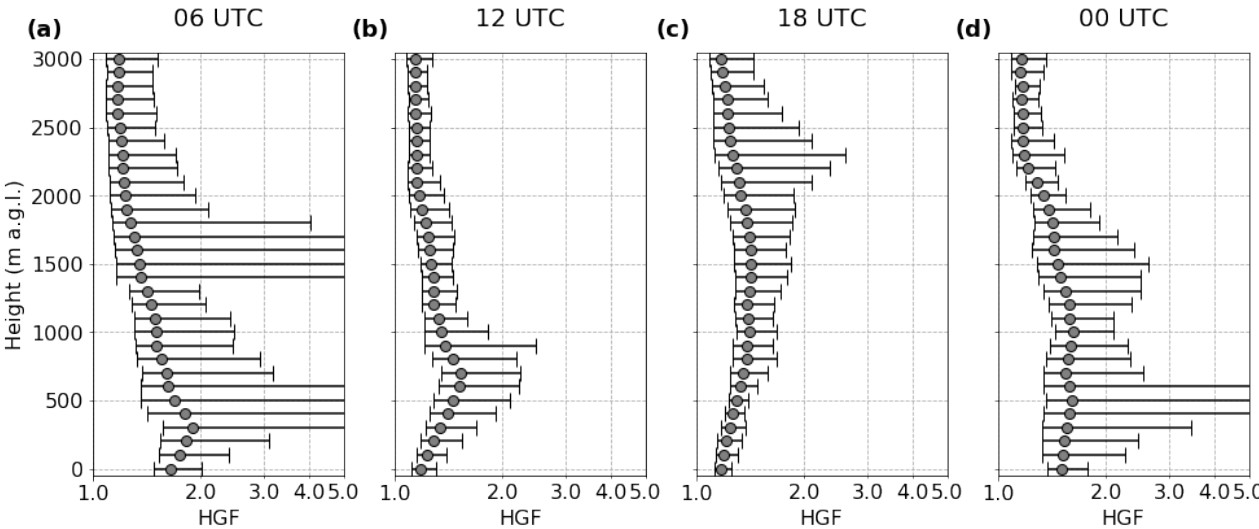

**Figure 5.** The mean and standard deviation of estimated hygroscopic growth factors at **(a)** 06 UTC, **(b)** 12 UTC, **(c)** 18 UTC and **(d)** 00 UTC. Where the upper value cannot be seen, this is where the air became saturated.

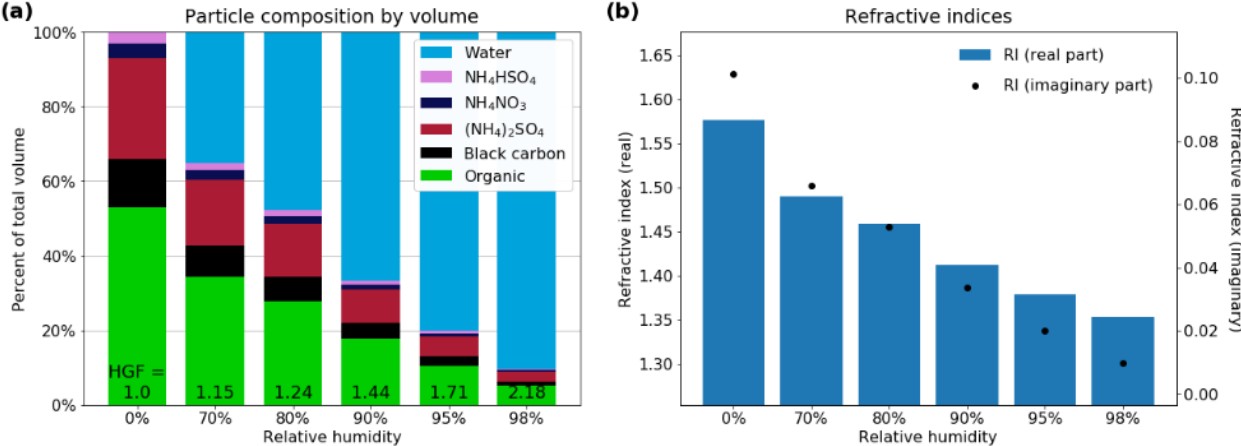

**Figure 6. (a)** The fractional composition of particles at different relative humidities and **(b)** the effect of the additional water volume on the particle's refractive index. These values have been calculated using the ZSR mixing rule.

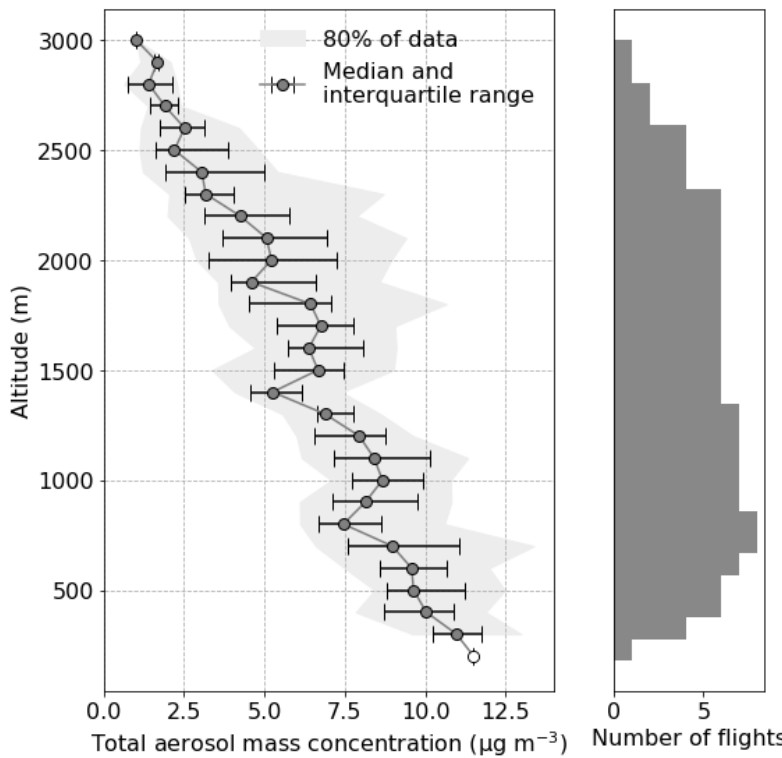

**Figure 7.** The average aerosol concentration profile from the area of interest. Circles and bars represent the median and the interquartile range, and the grey shading represents all data between the 10th and 90th percentiles. No flights went below 300 m in the dataset used here, so the open circle at 200 m is an estimation based on the median of data taken from flights further south. The histogram to the right shows the number of flights contributing data to each bin.

is little variation in the size and shape of the accumulation mode across different altitudes. The aerosol frequencies at different diameters generally fall within around 40 % of one another. This strongly suggests that a reasonably consistent accumulation mode can be expected, which will likely scale with the overall aerosol mass. In order to test the assumption that the Aitken mode does not contribute significantly to the AOD, the relative contribution of the average Aitken mode aerosol was calculated for a test case. The increase in AOD due to the addition of the Aitken mode aerosol was approximately 2.5 %. Given its limited contribution, only the accumulation mode aerosol was considered here.

The thick black lines in Fig. 3 show lognormal distributions that have been fitted to the accumulation mode. The mode is at 200 nm. The shape and mode of the distribution were assumed to remain constant across all altitude bins and the amplitude was adjusted according to the median aerosol mass at a given altitude. This produced an idealised accumulation mode size distribution of the same shape, but differing amplitudes at each altitude.

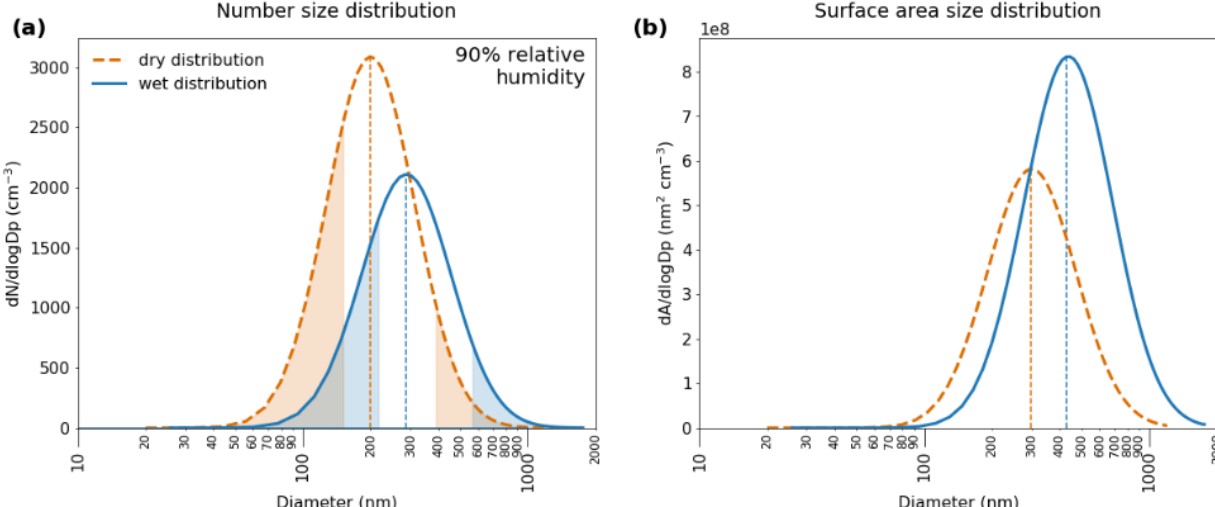

**Figure 8.** The dry and wet idealised **(a)** number and **(b)** volume size distributions based on an aerosol mass of $10\,\mu\mathrm{g\,m^{-3}}$ and an RH of $90\,\%$.

3. Grow the wet aerosol size distribution.

At each altitude bin, HGFs were used to 'grow' the dry size distribution (see Zieger et al., 2013). An HGF was established
and applied to each bin in the idealised dry size distribution, which allowed the shape of the new, wet size distribution
to be calculated. An example can be seen in Fig. 8 for an idealised dry size distribution based on an aerosol mass of
$10\,\mu\mathrm{g\,m^{-3}}$, assuming an RH of $90\,\%$.

4. Calculate AOD.

The chemical distribution, including water, of aerosol particles in each bin of the wet size distribution was used to cal-
culate particles' refractive indices. Their extinction coefficients were then calculated at $\lambda = 525$ nm using Mie code.
The total extinction coefficient in $\mathrm{Mm^{-1}}$ at each altitude was determined by integrating these across the whole wet size
distribution. This provides a quantification of the extent to which incoming radiation would be attenuated at each alti-
tude. Integrating these with altitude gives the AOD of the column. In Fig. 9, the total AOD for the example radiosonde
RH profile and the representative aerosol concentration are shown along with a profile of the dry and wet extinction
coefficients at each altitude. The total AOD of the wet profile in this case is approximately $60\,\%$ larger than that of the
dry profile.

This example shows the disproportionate contribution towards AOD from layers of high humidity, such as that high-
lighted in grey in this example. Although the wet extinction coefficients are higher than the dry ones at every altitude,
this effect is exaggerated in this humid layer.

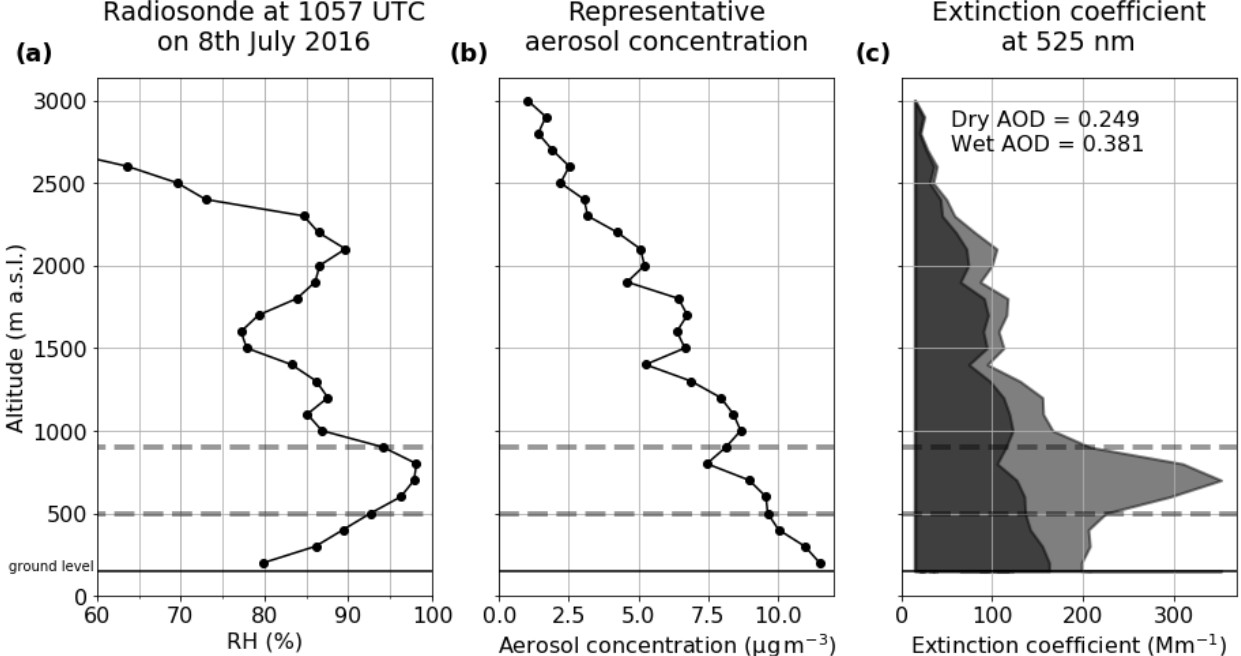

**Figure 9.** The **(a)** relative humidity and **(b)** aerosol concentrations used to calculate AOD, and **(c)** the profile of dry and wet calculated extinction coefficients for the case study RH profile example on 8 July. The solid black line shows ground level at Savè. The dashed grey lines outline the humid layer with RH greater than 95 %. In the third panel, the darker filled shape shows the dry extinction coefficient in the column and the lighter filled shape shows the wet extinction coefficient.

The approach outlined above has been used to calculate the AOD for all 63 unsaturated radiosonde RH profiles collected from Savè. Figure 10 shows a frequency plot of these calculated AODs, which are coloured according to the time of day the sonde was released (± one hour). These values are calculated purely based on variations in RH. The frequency distribution
therefore quantifies an estimate of the variation in AOD due entirely to RH; it does not show a true estimate of AOD for each profile, which would require a larger number of varying parameters. This plot is shown alongside a frequency distribution of the AODs measured by sun photometer from the Savè ground site. The sun photometer dataset includes 319 measurements taken between 8 June and 29 July 2016 as part of the DACCIWA field campaign.

There is good general agreement between the AOD values calculated here and those collected by sun photometer from the Savè groundsite over a similar period. The median AOD is 0.36 for both datasets. Values of AOD in the early morning have a larger spread, being reasonably well distributed between 0.3 and 0.7 in both cases, while AOD later in the evening in both cases was distributed more normally around the median. Both datasets have a high tail that is dominated by data from the mornings and the highest value is around 0.7 in the calculated dataset and 0.8 from the sun photometers. The assumption of a constant
aerosol profile prevents the model from capturing the full spread that would result from variations in aerosol loadings. As a

result, the sun photometer dataset has a larger interquartile range of 0.14, compared with 0.09 for the calculated dataset. The minimum value in the sun photometer dataset was a little less than 0.2, while that of the calculated dataset was closer to 0.3. This likely represents days with lower aerosol loadings, which are not captured by the model. The calculated dataset is based solely on observations in the lowest 3 km, and as such, any high-level aerosol layers are not taken into account. However, this does not appear to have significantly biased the calculated AODs.

It is important to note that the calculated values here are not expected to reproduce the real-world AOD for any given RH profile. A number of values are assumed to be constant here, including the shape and magnitude of the aerosol profile, the aerosol chemical composition, the shape of the accumulation mode size distribution and simplifications generally assumed in Mie calculations, such as that particles are internally-mixed spheres. A realistic calculation would require variation in such quantities to be taken into account. This dataset, instead, gives an estimate of the variability in the AOD measurements that is due entirely to changes in RH. Despite this limitation, this approach is able to replicate a large part of the measured AOD frequency distribution; in particular, it is able to reproduce the high values of AOD observed during the mornings. While the inclusion of aerosol variation would be necessary in order to recreate the full spectrum of AODs, there are no observations of aerosol concentrations from anywhere across the region that are sufficiently high alone to cause the enhancements in the ambient AOD observed by sun photometers. Although the omission of variations in aerosol properties has resulted in a narrower spread of calculated values compared with direct measurements, these results do show that the highly humid layers are the root cause of the very highest AODs being observed during the mornings.

## 3.3 The effect of high humidity layers on AOD

In the example case study shown in Fig. 9, the humid layer between 500 m and 900 m had a significant effect on the calculated column AOD. In Fig. 11, the relationship between highly humid layers and AOD is explored further. The maximum RH below 3 km was used as a simple parameter to quantify the presence of humid layers and the level of humidity within them for any given profile. The calculated AOD has been plotted here as a function of dry AOD ($f(\mathrm{AOD_{dry}}) = \mathrm{AOD_{wet}}/\mathrm{AOD_{dry}}$) against the maximum RH for each of the 63 unsaturated RH profiles in Fig.11. The $f(\mathrm{AOD_{dry}})$ represents how much larger the calculated wet AOD is than the dry AOD, so a value of 1 is equal to the dry AOD. This figure clearly shows an increase in wet AOD above dry in all cases. However, when layers of extremely high RH are present, this effect becomes more pronounced. When humid layers with RH higher than 97 % are present, the wet AOD is more than 1.5 times larger than the dry AOD; with layers of RH greater than 98 %, the wet AOD can be almost two times greater or more than the dry. The use of $f(\mathrm{AOD_{dry}})$ allows the potential for this value to be scaled according to the dry AOD from any aerosol profile.

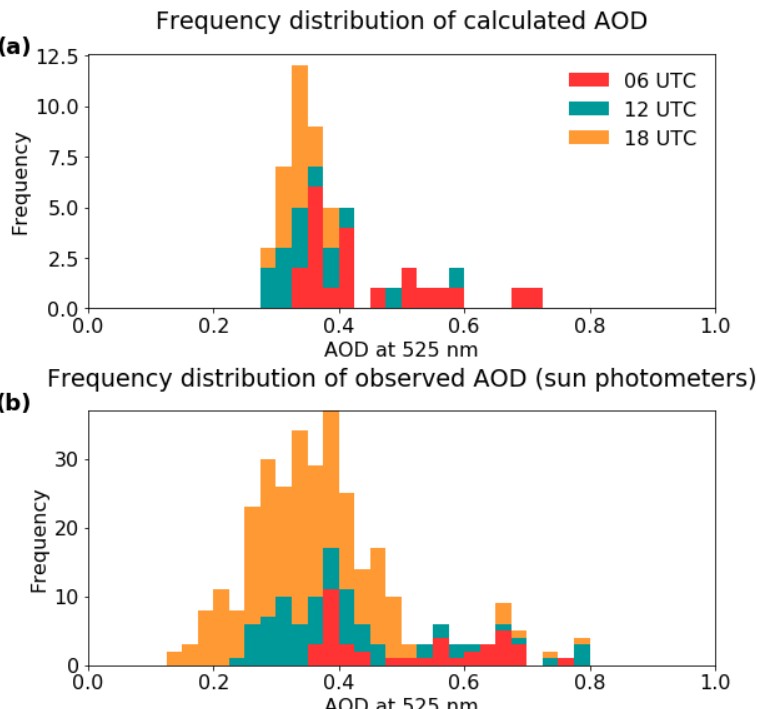

**Figure 10.** The frequency distributions of calculated AODs from radiosonde RHs, and AODs observed by sun photometers over Savé during the DACCIWA campaign. Bars shown here are stacked. Values have not been calculated for 00 UTC as this is during the night.

A curve of best fit has been added to Fig. 11 using Igor Pro's iterative least-squares algorithm. The resulting formula can be used to calculate an estimate of the $f(\mathrm{AOD_{dry}})$ from the dry AOD and the maximum RH in the profile (Eq. 5):

$$f(AOD_{dry}) = AOD_{dry} \left[ 1.25 + 1.3 \times \exp\left( \frac{RH_{max} - 100}{1.5} \right) \right] \tag{5}$$

where $RH$ is input as a value between 0 and 100. This is unlikely to be effective for values of $RH_{max}$ significantly lower than those explored here. The factor of 1.25, which implies an $f(\mathrm{AOD_{dry}})$ greater than the dry AOD when RH = 0, is likely an artefact created by the use of only the maximum value of RH for each RH profile. This empirical relationship could be useful for models that have no detailed treatment of aerosol chemistry.

The frequency distribution displayed in Fig. 11 shows that the RH reaches these high values a significant proportion of the time. Of the RH profiles gathered from Savé, 23 % included layers below 3 km that were unsaturated but contained layers of greater than 98 % humidity. This represents 37 % of the total number of unsaturated profiles.

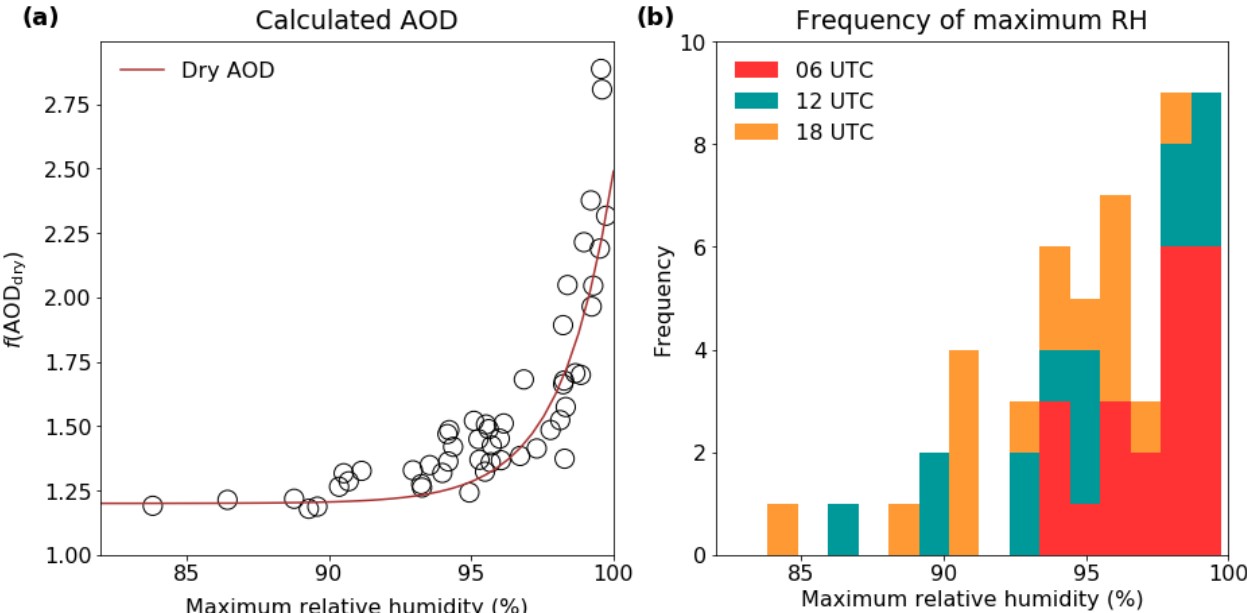

**Figure 11. (a)** The relationship between calculated AOD and the maximum sub-3 km RH in a profile, and **(b)** the frequency distribution of maximum unsaturated sub-3 km RH values. Bars in **(b)** are stacked.

A model-based study of aerosol liquid water content during the DACCIWA field campaign, carried out by Deetz et al. (2018a), found a mean dry AOD of 0.2 in southern West Africa, which increased to 0.7 when hygroscopic growth was taken into account. This mean dry aerosol is comparable with the dry AOD calculated here (0.25). The increase in AOD due to hygroscopic growth found here was slightly less: the median AOD calculated here was 0.36 (mean = 0.39) and the median of the measured AOD from the sun photometers was 0.36 (mean = 0.38). The study by Deetz et al. (2018a) considered a larger area, including regions further south and closer to the sea, which could explain the higher modelled hygroscopic growths in that study.

### 3.4 The effects of aerosol variability

Aerosol profiles have been assumed to be identical in all cases here. However, these loadings will in reality vary from day to day, which will have a resulting impact on the AOD. The extent of this impact can be estimated using the interquartile range of the aerosol measurements in the region, which are shown in Fig. 2. At all altitudes considered here, the interquartile range of the total aerosol loading was within $0.5\,\mu\mathrm{g\,m^{-3}}$ of $3.4\,\mu\mathrm{g\,m^{-3}}$. This is on average around a 27 % increase or decrease in the total aerosol population in less or more polluted circumstances, which would result in a similar change in the final calculated AOD. The median calculated AOD was 0.36; in this case, the less and more polluted circumstances would result in calculated AOD values of 0.26 and 0.46 respectively. This is a significant alteration in the final value and explains the broader range of

AODs observed by the sun photometers.


The AOD scales directly with the aerosol loading, but scales exponentially with increases in RH. Therefore, the effects of a humid layer on a day that is more polluted than average could have an even larger effect that those shown here. The highest AOD values calculated in this study are around 0.7; this could scale to almost 0.9 if a day is among the 25 % most polluted. As anthropogenic emissions in the region increase, the number of days reaching these very high AOD values will increase.

Furthermore, increased industrialisation can result in a higher proportion of inorganic, highly hygroscopic aerosol particles being produced (e.g. Morgan et al., 2009), which will exacerbate these effects.

These results suggest that the impact of highly humid layers on AOD, especially on days with high aerosol loadings, could be similar to that of low-level cloud. Capturing the presence of cloud in the region using satellites and other remote sensing

techniques is known to be difficult in the southern West African region; the detection of these sub-saturated humid layers adds an additional challenge. Nonetheless, it is important for their presence to be taken into account if the radiation balance in the region is to be fully understood.

Due to the irregular nature of aircraft data and the focus of this study on the impacts of RH, a number of limitations to the

scope of this study must be acknowledged. First, results shown here are based on observations of aerosol from the lowest 3 km of the atmosphere. In reality, some aerosol will exist at higher altitudes. For example, intrusions of biomass burning are known to become advected into this region regularly between 3 and 4 km at this time of year (Mari et al., 2008). Second, the focus on accumulation mode aerosol necessarily excludes the impact from other size modes from the calculations. Although very little coarse mode aerosol was observed by the Twin Otter aircraft during the DACCIWA campaign, it must be noted that any

present aerosol in this mode would increase the region's AOD. The Twin Otter flew above 2750 m only once, but at this time, a higher concentration of sulphate aerosol was observed that this altitude. If this datapoint is representative of the altitude in general, then it is likely that this study has underestimated the HGF at the higher altitudes. All of these factors together suggest that the AOD calculations provided here are likely to represent a lower limit.

## 4   Summary and conclusions

Aircraft and radiosonde data collected during the DACCIWA field campaign were used to estimate the hygroscopic growth of aerosols under different RH conditions, based on aerosol chemical composition and using the ZSR mixing rule. A consistent aerosol composition and vertical profile was assumed across the region of interest, due to the relative homogeneity of aerosol observations from the Twin Otter aircraft in the region. The RH profiles displayed a diurnal cycle, with higher RH generally observed during the night and early morning and drier conditions in the afternoon. These are features associated with the

Gulf of Guinea maritime inflow, described by Adler et al. (2017) and Deetz et al. (2018b), which brings cool air inland in the evening (Adler et al., 2018; Babić et al., 2018). Day-to-day variability was influenced by the different phases of the West

African Monsoon (Knippertz et al., 2017), which resulted in slightly more frequent high RH values during the first few weeks of July.

By assuming a constant aerosol composition and vertical profile, it was possible to explore the effects of variations in RH on the optical properties of aerosol in the region. Dry aerosol size distributions were grown according to the HGF, which had been calculated from the RH and the average chemical composition. Mie code was then used to calculate extinction coefficients at $\lambda = 525$ nm. This allowed AODs to be calculated for each of 63 unsaturated RH profiles.

The median AOD resulting from these calculations was 0.36. This compares with a median AOD of 0.36 found from observations at Savè using sun photometers. The shape of the AOD frequency distribution from the calculations was comparable to that from sun photometer measurements, although there was a greater spread in the sun photometer data due to variations in aerosol concentration. It was shown that the modest variations in aerosol properties observed cannot give rise to the elevated AOD alone. It is the large variations in RH that are required to account for many of the higher AOD measurements.

The calculated dry AOD using the representative aerosol profile was 0.25. In all cases, the wet AOD was found to be at least 25 % greater than the AOD for dry particles, showing the substantial impact of hygroscopic growth on optical properties. The rate of change of AOD with changes in RH was greater at higher RH. In particular, at values of RH greater than 98 %, the wet AOD was more than twice as large as the dry AOD. This scenario ocurred regularly in the field RH measurements, with 37 %

of unsaturated profiles featuring a humid layer of at least 98 % RH. These results are consistent with model results presented by Deetz et al. (2018a), who found that hygroscopic growth increased the AOD in the southern West African region from a dry value of 0.2 to a wet value of 0.7.

While aerosol loadings were assumed here to be constant, aircraft measurements in fact showed an interquartile range of

±27 %. Taking this variation into account would result in the average AOD varying between 0.26 and 0.46. It is important to note that the increase in AOD due to an increase in aerosol loading is linear, while that resulting from an increase in RH is exponential. Thus, the presence of highly humid layers in the column will substantially enhance any effect from increased aerosol loading. The highest AOD values would not be possible without this RH enhancement. Satellite-based retrievals of surface aerosol concentrations might thus be highly biased when humid layers are present.

These results show that the presence of highly humid layers during the monsoon season has a substantial impact on the direct radiative effect of aerosols in the southern West African region. Furthermore, as anthropogenic emissions from large cities in coastal West Africa increase and industrialisation increases the inorganic fraction, these humid layers will serve to amplify their effects. While detecting these layers and quantifying their effects on an ongoing basis is likely to provide a significant

challenge for the research community, results shown here suggest that this will be necessary if the radiation balance in the region is to be understood.

*Data availability.* All data used in this study are publicly available from the SErvice de DOnnées de l'OMP (SEDOO) database (http://baobab.sedoo.fr/DACCIWA).

*Author contributions.* SLH, JWT, KB, NK, AW, CD, FL, JB, RD and AS were involved in the collection, processing and
analysis of aircraft and ground site data during the DACCIWA campaign. SLH carried out the analysis for this study, with significant input from JWT, KD, BV, PZ and HC. The manuscript was prepared by SLH and all authors contributed to discussion and revision of the manuscript.

*Competing interests.* The authors declare that they have no conflict of interest.


*Special issue statement.* This article is part of the special issue *Results of the project "Dynamics–aerosol–chemistry–cloud interactions in West Africa" (DACCIWA)*

*Acknowledgements.* The research leading to these results has received funding from the European Union 7th Framework Programme (FP7/2007-2013) under Grant Agreement no. 603502 (EU project DACCIWA: Dynamics-aerosol-chemistry-cloud interactions in West
Africa). The lead author was supported by the NERC Doctoral Training Programme (grant ref: NE/L002469/1). Thanks to the German Weather Service (DWD) for providing access to the ICON forecast data. We thank British Antarctic Survey (BAS, operator of the Twin Otter aircraft), the Service des Avions Français Instrumentés pour la Recherche en Environnement (SAFIRE) and the Deutsches Zentrum für Luft- und Raumfahrt for their support during the aircraft campaign.

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
