# Peer review of "The radiative impact of out-of-cloud aerosol hygroscopic growth during the summer monsoon in southern West Africa"

_Atmospheric Chemistry and Physics, 2018_

## Referee Comment (RC1) · Anonymous Referee #1 · 9 Oct 2018

Review of "The radiative impact of out-of-cloud aerosol hygroscopic growth during the summer monsoon in southern West Africa"

This publication discusses the relationship between aerosol optical depth (AOD) and relative humidity. The study presents aircraft observations over western Africa of sub-micron aerosol size distributions and chemical composition along with observations of RH from aircraft and radiosondes. The authors use these measurements to estimate typical hygroscopic growth factors and AOD. Overall, the paper is clearly written and the authors note the strengths and limitations of the analysis. The measurements and analysis are interesting and of use to the aerosol and climate community. After

addressing a few minor comments, I would recommend publication in ACP.

1. The AOD calculations included here are only for sub-micron aerosol mass. From the representative the distributions in Figure 8 and Figure 3, there appears to be a fair amount of mass greater than 1 micron. I think some discussion of the limitation in size range and how this affects the AOD calculations should be included. Would a possible coarse mode composed of dust likely have a lower HGF than the accumulation mode (assuming dust to be hydrophobic)? How does this affect the AOD variability due to RH?

2. The authors assume the chemical composition measured by the AMS to be constant across the size distribution from the SMPS (20 - 500 nm). How valid is this assumption and how sensitive are the results? If the inorganic species contribute more to the Aitken mode peak and the organics to the accumulation mode peak, does this not reduce the calculated HGF?

3. In Figure 2, are the black carbon concentrations from the SP2 (and not the AMS)?

4. The interquartile range is used to provide an assessment of typical variability in aerosol concentration, but does this capture very high aerosol episodes (biomass burning, dust) that may impact West Africa? I suggest including a brief statement on this.

5. Why was only the accumulation mode used in the calculation of AOD? From Figure 3b, it seems there is some volume missing from the fitted-accumulation-mode distribution.

6. As a small stylistic suggestion, the first part of Section 3 could be a sub-section (for instance, 3.1 Observations). I thought the measurements of aerosol properties and RH was an interesting part of the study. Including it under a sub-section could help highlight this.

---

## Referee Comment (RC2) · Anonymous Referee #2 · 12 Oct 2018

This manuscript describes measurements and estimates of particle water during a field campaign (DACCIWA) in West Africa in June and July 2016. The investigators present measurements of particle mass from an AMS aboard aircraft platforms and RH measurements from balloon launches. The authors identify an important open question in atmospheric chemistry, that is, what is the concentration of particle water and how can that be used to understand the radiative impacts of atmospheric particulate matter. There is literature that relates particle water to AOD, and it is not cited here. I find the manuscript tries to make broad statements beyond their analysis. As case studies this manuscript could be very good. I cannot recommend the article for publication in its current form.

[Figure]

Specific comments

Abstract: Line 1: atmospheric water can also exist in the solid phase The authors state: "at high humidities more water vapor condenses onto particles.." This is true for a constant particle concentration and unchanging chemical composition. A main conclusion in the abstract is "Therefore, …. AOD … can be described by relative humidity alone." The evidence presented here does not support that conclusion and contradicts other published manuscripts from the same campaign.

Page 2, starting at Line 29: the extent to which particles take on water is also dependent on the particle concentration in addition to chemical composition.

Page 3, Line 58: I find a lack of support that ZSR calculations are more reliable than HGF calculations for nitrate-containing particles. Hennigan et al., ACP, 2015 (doi:10.5194/acp-15-2775-2015) and Guo et al., ACP, 2015 (doi:10.5194/acp-15-5211-2015) find closure when considering nitrate. Perhaps the calculations here may be more accurate, but the "likely more reliable" statement is not well supported.

Page 3, Line 65: The authors should back up their statement that organic compounds contribute less to particle water than inorganic species. There are several examples. Since the investigators used an AMS: Nguyen et al., ES&TL, 2016 (DOI: 10.1021/acs.estlett.6b00167).

Page 3, Line 82: The basic premise of this manuscript is that HGF is dependent on RH not particle concentration or chemical composition, which directly contradicts the argument the authors make here, that b/c anthropogenic emissions are expected to increase, hygroscopic growth will be impacted. They could argue RH might change … but they should not contradict themselves.

Page 4, sentence beginning at line 107: The authors state mineral dust contributes little to aerosol volume. This assertion is not well defended and is contradicted in a recent manuscript: "Potential climate effect of mineral aerosols over West Africa:

Part II—contribution of dust and land cover to future climate change: by Ji et al. https://link.springer.com/article/10.1007/s00382-015-2792-x. Would the authors resolve this statement in the context of the existing literature?

Page 8, Starting at line175: The authors could not discern a diurnal profile due to insufficient sampling and then 'therefore' assume constant chemical composition throughout the day is not well defended. Figure 7 from the campaign's overview paper "THE DYNAMICS–AEROSOL–CHEMISTRY–CLOUD INTERACTIONS IN WEST AFRICA FIELD CAMPAIGN" Flamant et al., BAMS, 2018 shows a time-of-day change in aerosol backscatter (related to HGF). It is rises during the day and is not lowest when RH is lowest. Likely this observation is due to changing aerosol concentration and chemistry, and subsequently kappa and properties that change particle growth factors are changing. If RH is the predominant controlling factor, why does the timing of Flamant's Figure 7 of backscatter not match the time profiles of RH here? I cannot accept the stated assumption as a 'therefore'.

Figure 7: how are data extrapolated above ∼2500 asl specifically? Is it a linear extrapolation? Typically these profiles are assymptotic to zero. Instead of plotting an example profile, it's my opinion that a distribution about the mean or median would be better. I think it unlikely the vertical profiles all match the average so well, but acknowledge there may be little variability and I may be wrong. As presented it is difficult to tell. Also, is this aerosol mass as measured by the AMS or do the estimates include particle water?

Figure 1: The authors consider data from only a small fraction of this map. The excluded areas should be masked in some way to highlight they are using a subset of data that is not representative of this entire map.

Page 7, Line 154: Can the authors back up how they know the aerosol was "acidically neutral" in all studied cases.

Figure 2: It seems the authors flew to ∼3000 a.s.l .but only present data to 2000 a.s.l.

Why? The nitrate-to-carbon ratio changes from a factor of ∼5 to 2.5. Would the authors explain what they mean precisely by "stable chemical distribution" *line 170.

Page 8, Line 175: The authors state that because they could not attain a sufficient sample size ... "therefore" the aerosol concentration and distribution is ... constant throughout the day. This does not logically follow and contradicts data presented in manuscripts from this campaign, and I would argue data presented here.

Figure 10: At first I thought this distributions were over-layed and then after seeing Figure 11 it seems they are stacked. This is not clear. What is the physical meaning of AOD>1? It seems the authors use a qualitative visual inspection of a) and b) (which have different y-axis limits) to state definitively similarity. I do not think this conclusion is well-founded.

Editorial I find the introduction disjointed and in many way disconnected from the manuscript's science. It seems sections were written by different people in different styles and the manuscript does not read as a coherent document.

---

## Author Comment (AC1) · 16 Dec 2018

Many thanks the the reviewers for their comments and suggestions. Please find attached our response to the reviewer comments and an updated version of the manuscript with changes highlighted.

Please also note the supplement to this comment: https://www.atmos-chem-phys-discuss.net/acp-2018-805/acp-2018-805-AC1-supplement.pdf

2018.

---

## Author Response (AR1)

**Response to reviewers for "The radiative impact of out-of-cloud aerosol hygroscopic growth during the summer monsoon in southern West Africa" *by* Sophie L. Haslett et al.**

We thank the two anonymous reviewers for their useful comments and suggestions.

During the process of revisiting the work leading to this paper, an error in the code involved in growing the wet aerosol size distributions was discovered. This has been corrected. Although it has led to some changes in the data presented in Figs. 8, 9, 10 and 11, the overall message of the study has remained the same. Paul Zieger has been added to this paper as a co-author to acknowledge his contribution to discussions related to these corrections.

Reviewers' comments are shown here in bold font and our responses in red.

**Anonymous Referee #1**

**This publication discusses the relationship between aerosol optical depth (AOD) and relative humidity. The study presents aircraft observations over western Africa of submicron aerosol size distributions and chemical composition along with observations of RH from aircraft and radiosondes. The authors use these measurements to estimate typical hygroscopic growth factors and AOD. Overall, the paper is clearly written and the authors note the strengths and limitations of the analysis. The measurements and analysis are interesting and of use to the aerosol and climate community. After addressing a few minor comments, I would recommend publication in ACP.**

**1. The AOD calculations included here are only for sub-micron aerosol mass. From the representative the distributions in Figure 8 and Figure 3, there appears to be a fair amount of mass greater than 1 micron. I think some discussion of the limitation in size range and how this affects the AOD calculations should be included. Would a possible coarse mode composed of dust likely have a lower HGF than the accumulation mode (assuming dust to be hydrophobic)? How does this affect the AOD variability due to RH?**

Figure 3 shows the dry size distribution measured by the SMPS, which was only able to measure up to 0.5 µm (500 nm), so aerosol greater than 1µm cannot be seen in this case. If the fitted lognormal curve is extended above 0.5 µm, however, as it is in the dashed yellow curve shown in Fig. 8, this still shows the majority of the dry aerosol distribution falling below 1 µm. This fitted lognormal was used for calculations and to calculate the wet aerosol distribution, so even in cases where the wet aerosol distribution was significantly larger, these > 1µm values were still included in the calculations.

While the SMPS cannot measure coarse mode particles, there was a Grimm optical particle counter present on the Twin Otter aircraft during the DACCIWA campaign, which was able to observe the coarse mode. Very little coarse mode aerosol was seen during this campaign, however. This is likely due to the monsoon winds bringing air into the region from the south at this time of year, which led to very little Saharan dust being present in the lower atmosphere. An acknowledgement of this has been included in the manuscript in line **198** and a further acknowledgements of related limitations in this study have been included in lines **404-413**. Given the weak hygroscopicity of Saharan dust, if there were a

significant coarse mode, this would increase the dry AOD more than the wet, and thus decrease the difference between them.

**2. The authors assume the chemical composition measured by the AMS to be constant across the size distribution from the SMPS (20 - 500 nm). How valid is this assumption and how sensitive are the results? If the inorganic species contribute more to the Aitken mode peak and the organics to the accumulation mode peak, does this not reduce the calculated HGF?**

The AMS measures the aerosol mass for each of the different species, which is much more concentrated in the accumulation mode than the number distribution, even in cases where the Aitken mode is large. This can be seen by the volume size distribution shown below, which is directly proportional to the aerosol mass. Therefore, even if our assumption is incorrect and the Aitken mode composition is different to the accumulation mode, the chemical composition measured by the AMS will be strongly biased towards the accumulation mode concentration. We feel this makes the use of the AMS chemical composition a fair assumption in this case, given that our focus is on the accumulation mode and our calculations of accumulation mode composition will be accurate to within 10%.

[Figure]

**3. In Figure 2, are the black carbon concentrations from the SP2 (and not the AMS)?**

Yes - this has now been updated in the figure caption. Thank you for spotting this.

**4. The interquartile range is used to provide an assessment of typical variability in aerosol concentration, but does this capture very high aerosol episodes (biomass burning, dust) that may impact West Africa? I suggest including a brief statement on this.**

We agree with the referee and have included a discussion of this limitation in lines **404-413**.

**5. Why was only the accumulation mode used in the calculation of AOD? From Figure 3b, it seems there is some volume missing from the fitted-accumulation-mode distribution.**

The accumulation mode was found to contain the most optically active aerosol; we tested the influence on the AOD of including the Aitken mode and it was found to contribute only around 2.5% to the AOD. A description of this has now been added to lines **288-291**.

**6. As a small stylistic suggestion, the first part of Section 3 could be a sub-section (for instance, 3.1 Observations). I thought the measurements of aerosol properties and RH was an interesting part of the study. Including it under a sub-section could help highlight this**

Thank you for this suggestion – this subtitle has now been added to the beginning of the results section.
* * *
**Anonymous Referee #2**

**This manuscript describes measurements and estimates of particle water during a field campaign (DACCIWA) in West Africa in June and July 2016. The investigators present measurements of particle mass from an AMS aboard aircraft platforms and RH measurements from balloon launches. The authors identify an important open question in atmospheric chemistry, that is, what is the concentration of particle water and how can that be used to understand the radiative impacts of atmospheric particulate matter. There is literature that relates particle water to AOD, and it is not cited here. I find the manuscript tries to make broad statements beyond their analysis. As case studies this manuscript could be very good. I cannot recommend the article for publication in its current form.**

The referee's main criticism with our paper seems to be that we are assuming that relative humidity is the sole driver of variations in AOD and aerosol number and composition does not play a role. This is a misinterpretation of the paper. We sought to show that in West Africa, since aerosol properties are rather invariant and relative humidity variations are large and that since AOD is linearly dependent on aerosol number but exponentially dependent on relative humidity, it is the high humidity that is important in driving up the absolute AOD, and this is only moderated by variation in aerosol. We also recognize that as aerosol pollution increases the instances of high AOD will become even greater. We thank the referee for their comments since they have highlighted the lack of clarity in our initial paper and have greatly helped to improve our revision. In addition, the referee's comments have led to us interrogating our data and identifying an error in our earlier calculations and we are very grateful for this.

**Specific comments**

**Abstract: Line 1: atmospheric water can also exist in the solid phase The authors state: "at high humidities more water vapor condenses onto particles.." This is true for a constant particle concentration and unchanging chemical composition.**

These details have now been added to the abstract in lines **1-2**.

**A main conclusion in the abstract is "Therefore, . . .. AOD . . . can be described by relative humidity alone." The evidence presented here does not support that conclusion and contradicts other published manuscripts from the same campaign.**

Our point was not to argue that variations in the abundance or composition of aerosol does not have an effect on hygroscopicity but rather that the observations of elevated AOD can only be explained by high humidities at the top of the boundary layer given the lack of variation in aerosol loading and composition observed throughout the region. The referee seems to have misunderstood the point we

were trying to make, and we acknowledge that this is perhaps we were not as clear as we should have been. We have rephrased the abstract to try and remove any ambiguity.

**Page 2, starting at Line 29: the extent to which particles take on water is also dependent on the particle concentration in addition to chemical composition.**

This has been added to line **33**.

**Page 3, Line 58: I find a lack of support that ZSR calculations are more reliable than HGF calculations for nitrate-containing particles. Hennigan et al., ACP, 2015 (doi:10.5194/acp-15-2775-2015) and Guo et al., ACP, 2015 (doi:10.5194/acp-15-5211- 2015) find closure when considering nitrate. Perhaps the calculations here may be more accurate, but the "likely more reliable" statement is not well supported.**

The 'likely more reliable' phrasing has been removed and the two articles mentioned here have been included as examples where nitrate closure was achieved in lines **61-64**.

**Page 3, Line 65: The authors should back up their statement that organic compounds contribute less to particle water than inorganic species. There are several examples. Since the investigators used an AMS: Nguyen et al., ES&TL, 2016 (DOI: 10.1021/acs.estlett.6b00167).**

This has now been added to line **69**.

**Page 3, Line 82: The basic premise of this manuscript is that HGF is dependent on RH not particle concentration or chemical composition, which directly contradicts the argument the authors make here, that b/c anthropogenic emissions are expected to increase, hygroscopic growth will be impacted. They could argue RH might change . . . but they should not contradict themselves.**

That most certainly was not the intention of the paper. Referee 1 has recognized this, but we appreciated that we did not make ourselves clear enough in the initial paper and thank the referee for allowing us to make our premise more clearly in the revision.

The focus of this work is on the impacts of high humidity, but our intention was not to downplay the aerosol effects. In essence, our objective was to examine the impact of high humidity on the AOD. The climatology and pollution affecting West Africa means that the aerosol loading varies only moderately but the humidity varies widely. Our results show that the very highest AODs are only possible due to the presence of highly humid layers – ie, the high RH exacerbates the aerosol AOD. Since AOD responds linearly to increases in aerosol and exponentially to increases in RH, we show that high RH is a necessary condition to reproduce the highest AODs observed. As anthropogenic aerosol in the region increases, the severity of these high AODs will increase.

We have aimed to clarify this throughout the text, including in the abstract (lines **16-23**), in the results (lines **214-219**, **329-335** and **343-349**) and in the conclusion (lines **432-434**).

**Page 4, sentence beginning at line 107: The authors state mineral dust contributes little to aerosol volume. This assertion is not well defended and is contradicted in a recent manuscript: "Potential climate effect of mineral aerosols over West Africa: Part II Tcontribution of dust and land cover to future climate change: by Ji et al. ˇ https://link.springer.com/article/10.1007/s00382-015-2792-x. Would the authors resolve this statement in the context of the existing literature?**

This is an important point and has been explored in more detail now in lines **114-118**. While dust typically makes a significant contribution to West African aerosol, during the monsoon season in the southerly region that was explored during DACCIWA, the wind is predominantly from the south and dust has much less influence. It is a valuable addition to have this discussed now in the text.

**Page 8, Starting at line175: The authors could not discern a diurnal profile due to insufficient sampling and then 'therefore' assume constant chemical composition throughout the day is not well defended. Figure 7 from the campaign's overview paper "THE DYNAMICS–AEROSOL–CHEMISTRY–CLOUD INTERACTIONS IN WEST AFRICA FIELD CAMPAIGN" Flamant et al., BAMS, 2018 shows a time-of-day change in aerosol backscatter (related to HGF). It is rises during the day and is not lowest when RH is lowest. Likely this observation is due to changing aerosol concentration and chemistry, and subsequently kappa and properties that change particle growth factors are changing. If RH is the predominant controlling factor, why does the timing of Flamant's Figure 7 of backscatter not match the time profiles of RH here? I cannot accept the stated assumption as a 'therefore'.**

The backscatter from the ceilometer shown by Flamant et al. (2018) shows the cloud base height rising throughout the day; however, below the clouds, the backscatter can be seen to start high and decrease throughout the day, which is consistent with our measurements. Above the clouds, the RH in the Flamant et al. (2018) figure varies depending on whether there is a cloud present, which can be seen for example between 1400-1500. It is important to note that this figure is an example from one day and is not representative of the broad scale average.

Efforts have been made to explore the diurnal cycle based on the aircraft results, and very little variation was found. This has now been described and justified in much more detail in lines **201-212**, as we recognise that this was not well phrased in the original manuscript. Much of the accumulation aerosol in the region is thought to have originated from biomass burning in central/southern Africa and to have been the result of long-distance transport, which is a possible explanation for the limited variability seen here. This is the focus of a further manuscript that is about to be submitted to ACP. The limited diurnal variation is shown here:

[Figure]

**Figure 7: how are data extrapolated above ~2500 asl specifically? Is it a linear extrapolation? Typically these profiles are assymptotic to zero. Instead of plotting an example profile, it's my opinion that a distribution about the mean or median would be better. I think it unlikely the vertical profiles all match the average so well, but acknowledge there may be little variability and I may be wrong. As presented it is difficult to tell. Also, is this aerosol mass as measured by the AMS or do the estimates include particle water?**

After considering this point, the median profile has now been used for calculations in place of the original representative profile. The median, interquartile range and 10[th] and 90[th] percentiles have been

plotted in figure 9, alongside a histogram showing the number of flights that contributed data in each altitude bin. There was no available data in the target region at 200 m, so an average value from further south has been used in this case. The aerosol mass represented in this figure is dry aerosol mass – this has been added to the figure caption for clarity.

**Figure 1: The authors consider data from only a small fraction of this map. The excluded areas should be masked in some way to highlight they are using a subset of data that is not representative of this entire map.**

The map has been updated, with parts of the map not included in this study faded out to make the area used stand out.

**Page 7, Line 154: Can the authors back up how they know the aerosol was "acidically neutral" in all studied cases.**

This was a miss-phrasing and has been updated in line **164**.

**Figure 2: It seems the authors flew to ~3000 a.s.l .but only present data to 2000 a.s.l Why? The nitrate-to-carbon ratio changes from a factor of ~5 to 2.5. Would the authors explain what they mean precisely by "stable chemical distribution" *line 170.**

This was an oversight. Bins at 2500 m and 3000 m have now been added. The 3000 m bin has been faded in the figure, as it was produced from only one datapoint. The different chemical composition seen at this point and its potential impact on our results have been acknowledged in the text in lines **189-191** and lines **410-413**.

**Page 8, Line 175: The authors state that because they could not attain a sufficient sample size . . . "therefore" the aerosol concentration and distribution is . . . constant throughout the day. This does not logically follow and contradicts data presented in manuscripts from this campaign, and I would argue data presented here.**

This has been addressed above – a much more rigorous explanation of the reasoning leading to this decision has now been included in lines **201-212**.

**Figure 10: At first I thought this distributions were over-layed and then after seeing Figure 11 it seems they are stacked. This is not clear. What is the physical meaning of AOD>1? It seems the authors use a qualitative visual inspection of a) and b) (which have different y-axis limits) to state definitively similarity. I do not think this conclusion is well-founded.**

The figure caption has been updated to make it clear that the distributions are stacked.

With changes that have now been made to some of the calculations, the AOD calculations do not now exceed 1. There is, however, no physical upper limit to AOD, so values higher than 1 could have been possible.

After updates that have been made to the calculations, the calculated dataset now appears slightly different. Efforts have been made in the text to describe the similarities and differences between the two distributions more clearly, with possible reasons for both explored, in lines **325-335**, rather than simply stating similarity. The difference in y-axis values is purely down to the different number of

measurements: the y-axis represents frequency, and more sun photometer measurements were made over this period than there were radiosondes released.

**Additional references:**

[revised manuscript text omitted]